# Complex electronic structure and compositing effect in high performance thermoelectric BiCuSeO

Guang-Kun Ren [1,2,3], Shanyu Wang[2], Zhifang Zhou[1], Xin Li[4], Jiong Yang[4], Wenqing Zhang[4], Yuan-Hua Lin[1], Jihui Yang [2] & Ce-Wen Nan[1]

BiCuSeO oxyselenides are promising thermoelectric materials, yet further thermoelectric figure of merit ZT improvement is largely limited by the inferior electrical transport properties. The established literature on these materials shows only one power factor maximum upon carrier concentration optimization, which is typical for most thermoelectric semiconductors. Surprisingly, we found three power factor maxima when doping Bi with Pb. Based on our first-principles calculations, numerical modeling, and experimental investigation, we attribute the three maxima to the Fermi energy optimization, band convergence, and compositing effect due to in situ formed PbSe precipitates. Consequently, three ZT peaks of 0.9, 1.1, and 1.3 at 873 K are achieved for 4, 10, and 14 at.% Pb-doped samples, respectively, revealing the significance of complex electronic structure and multiple roles of Pb in BiCuSeO. The results establish an accurate band structure characterization for BiCuSeO and identify the role of band convergence and nanoprecipitation as the driving mechanism for high ZT.

---

[1] State Key Laboratory of New Ceramics and Fine Processing, School of Materials Science and Engineering, Tsinghua University, 100084 Beijing, China. [2] Materials Science and Engineering Department, University of Washington, Seattle, WA 98195, USA. [3] Institute of Materials, China Academy of Engineering Physics, Jiangyou 621908, China. [4] Materials Genome Institute, Shanghai University, 200333 Shanghai, China. Correspondence and requests for materials should be addressed to Y.-H.L. (email: linyh@mail.tsinghua.edu.cn) or to J.Y. (email: jihuiy@uw.edu)

With the increasing energy demands and environmental concerns, seeking renewable energy solutions and developing technologies with high energy conversion efficiency are of paramount significance[1]. Thermoelectric (TE) technology, capable of directly converting various waste heat into electricity based on the Seebeck effect, or providing electronic cooling based on the Peltier effect, could play an pivotal role in a sustainable future[2]. The attributes of TE devices, including excellent reliability, scalability, no moving part or emission, and so on, make them promising for recovering low-to-intermediate temperature (500–900 K) waste heat from industry sectors or vehicle exhaust[3,4]. The low conversion efficiency, scarcity of constituents (e.g., Te), and thus high cost of typical TE materials, such as $Bi_2Te_3$ and PbTe, and so on, however, largely limit the large-scale applications of TE technology. With the virtues of ultralow lattice thermal conductivity ($\kappa_L$ ~0.4 W m$^{-1}$ K$^{-1}$ for the pristine sample at high temperatures), high Seebeck coefficient, and good thermochemical stabilities, BiCuSeO is one of the most promising candidates for mid-temperature TE power generation applications[5–7]. High TE figures of merit (ZT) of 1.2–1.5 have been achieved in BiCuSeO by utilizing the strategies such as modulation doping[8], texturing[9], hierarchic structuring[10], chemical bonding engineering[11], and so on. As compared with typical TE semiconductors, for example, $Bi_2Te_3$[12] and PbTe[13], however, its relatively low power factor (PF) (normally <10 μW cm$^{-1}$ K$^{-2}$) largely hinders the further improvement of ZT. Recently, many approaches, including carrier concentration engineering[14,15], band structure engineering by modifying the band convergence[16,17] or introducing sharp features in the electronic density of states (DOS)[3], carrier scattering mechanism engineering[18], unique band features driven by the spin–orbit coupling (SOC)[19], and so on, emerge to largely improve the PF and thus ZT of TE materials.

The electronic band structure of BiCuSeO calculated by first-principles density functional theory (DFT) shows a complex constitution with multiple valleys near the valence band maximum (VBM), and the partial DOS plots further reveal that VBM is dominated by the hybridized Cu 3d and Se 4p orbitals[20]. These multiple conduction valleys could contribute simultaneously but weigh differently to the electrical transport at a specific Fermi energy ($E_F$), depending on the band effective mass and their relative positions to $E_F$. Inspired by the work done by Zhang et al.[21] in the $Mg_3Sb_2$-based system, a comprehensive, systematic investigation of the correlation between band features (e.g., DOS, band convergence) and carrier transport properties is required to fundamentally understand and further optimize the carrier transport of BiCuSeO, though its large Seebeck coefficients (>150 μV K$^{-1}$) and moderate PF values (~7 μW cm$^{-1}$ K$^{-2}$) are normally attributed to the complex band structure. In addition, the increasing band degeneracy ($N_v$) with down-shifting the $E_F$, and the evident band nonparabolicity could give rise to unique electrical properties, as compared to other typical TE materials showing only single PF and/or ZT peak upon carrier concentration modification. These will bring about additional challenges in studying the band-transport correlation, but do offer opportunities to optimize the electrical transport properties.

In this work, we have successfully prepared $Bi_{1-x}Pb_xCuSeO$ samples ($x = 0$–0.2) by a time- and energy-efficient self-propagating high-temperature synthesis and spark plasma sintering (SHS-SPS) technique, and systematically investigated their high-temperature TE properties and the correlation between the band structure features and electrical transport properties. With the $E_F$ shifting down, that is, increasing the carrier concentration by increasing the Pb content, two maxima in the PF and ZT values have been observed. The first one is traceable to the convergence of multiple valence bands, and the subsequent results from the conventional trade-off between the electrical conductivity ($\sigma$) and Seebeck coefficient ($S$). Additional PF and ZT maxima can be observed with further increase in the Pb content beyond the solubility limit ($x > 0.12$), primarily due to the compositing effect originated from in situ formed PbSe precipitates. Consequently, ZT peaks of 0.9, 1.1, and 1.3 at 873 K are achieved for 4, 10, and 14 at% Pb-doped BiCuSeO samples, respectively.

## Results

**Electronic band structure of BiCuSeO.** The electronic band structure and DOS of BiCuSeO were calculated taking into account the SOC effect and are shown in Fig. 1. The modified Becke–Johnson (mBJ) potential was adopted for band-gap correction, and the calculated indirect gap is ~0.8 eV, well consistent with the experimental results[22,23]. The conduction band maximum locates at the Z point, while the VBM lies along the $\Gamma$–M line. Besides, the VBM shows a large dispersion, indicating its light band feature, while multiple heavy valleys along the $\Gamma$–Z–R line exist below the VBM (within several $k_BT$, $k_B$ is the Boltzmann constant, as shown in the shaded area). The combination of multiple light and heavy bands could be beneficial for the electrical properties, as will be discussed in more details below. The complex valence band structure is also validated by the DOS plot, showing multiple sharp peaks at the valence band top (within 1.5 eV below the VBM). The first DOS peak at ~−0.2 eV, as highlighted in the shaded area, dominates the hole transport, considering the optimal carrier concentration (<10$^{22}$ cm$^{-3}$) for most TE semiconductors. The sharp DOS peak is mainly composed of the hybridized orbitals from Cu and Se, well consistent with the conductive functionality of $(Cu_2Se_2)^{2-}$ layers (Supplementary Fig. 2). Figure 1c shows the first Brillouin zone and Fermi surfaces plotted at different $E_F$ corresponding to the hole concentrations $p$ of $3.0 \times 10^{20}$, $5.5 \times 10^{20}$, and $1.2 \times 10^{21}$ cm$^{-3}$. Even for lightly doped samples with $E_F$ being just below the VBM, for example, $p = 3.0 \times 10^{20}$ cm$^{-3}$, the Fermi surface consists of multiple valleys, including dual degeneracy at the Z point, quadruplet along the $\Gamma$–M line, and eightfold degeneracy along the Z–R line. With down-shifting the $E_F$, these valleys contribute increasingly and simultaneously to the carrier conduction. The sharp DOS peak at ~−0.2 eV is primarily due to the incorporation of multiple heavy valleys with significant band nonparabolicity (along the $\Gamma$–Z–R line). The combination of light/heavy bands and high band degeneracy $N_v$ ~14 has been widely accepted as the origin of the high $S$ and modest PF of BiCuSeO, considering its low $\mu_H$ and thus low $\sigma$. A systematic investigation of the complex band structure, with the aim at fully utilizing the favorable band features, however, is missing. In the following part, $E_F$ is gradually moved down facilitated by Pb doping on the Bi site, and the electronic band structure, carrier transport properties, and their correlations are systematically explored.

**Exploring the complex band structure.** Phase compositions of the bulk samples were characterized by X-ray diffraction (XRD) (Supplementary Fig. 3a). The diffraction patterns can be well indexed to tetragonal BiCuSeO (PDF# 45-0296, Supplementary Fig. 3b) for all Pb-doped samples[11], and there is no detectable impurity phase for samples with the nominal Pb content up to 12 at%. For the $x > 0.12$ samples, three secondary peaks appear, which can be indexed to be cubic PbSe (PDF# 06-0354)[24]. Meanwhile, the (211) peak of $Cu_2Se_\delta$ (PDF# 47-1448)[25] is also detected in the $x > 0.12$ samples, and our previous work has confirmed that this impurity phase is mainly formed during the rapid SHS processes[11]. These impurity phases, as will be discussed below, show a large influence on the TE transport for the $x > 0.12$ samples. The calculated lattice parameters of

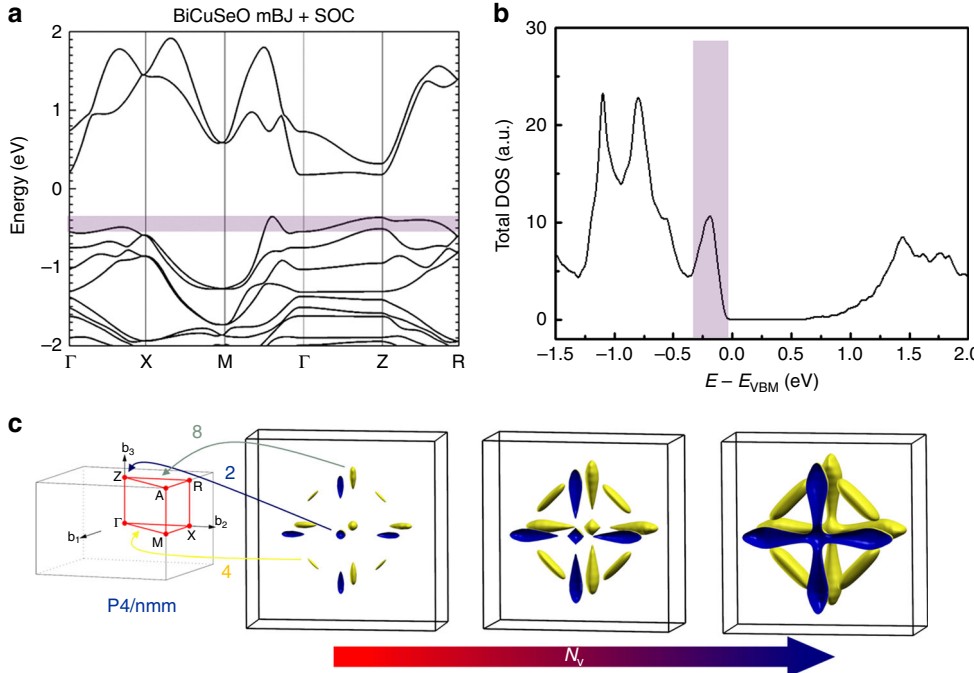

**Fig. 1** Electronic structure and Fermi surfaces of BiCuSeO. **a** Band structure of the pristine BiCuSeO. **b** Total electronic density of states (DOS). The shaded areas show the valence bands that dominate the hole transport. **c** The first Brillouin zone and Fermi surfaces at different carrier concentrations ($p = 3.0 \times 10^{20}$, $5.5 \times 10^{20}$, and $1.2 \times 10^{21}$ cm$^{-3}$)

$Bi_{1-x}Pb_xCuSeO$ ($x = 0$–0.2), shown in Supplementary Fig. 3c, indicate that the solubility limit of Pb in BiCuSeO is ~12 at%, consistent with the appearance of impurity phases for $x > 0.12$ samples in the XRD data.

Carrier concentration $p$ and Hall mobility $\mu_H$ values of the Pb-doped BiCuSeO at room temperature are shown in Fig. 2a and listed in Table 1. Pb doping on the Bi site successfully increases $p$, in agreement with the previous studies[23,26–28]. Compared with the pristine BiCuSeO, $p$ increases by three orders of magnitude and reaches ~$1.9 \times 10^{21}$ cm$^{-3}$ for $Bi_{0.88}Pb_{0.12}CuSeO$. The experimental $p$ values agree well with the theoretical predication considering Pb as an acceptor for $x \leq 0.12$; however, $p$ remains unchanged when the Pb content goes beyond the solubility limit. The results are in good agreement with the XRD and lattice parameter data, indicating excellent doping efficiency of Pb. The estimated $E_F$ plotted in Fig. 2b increases gradually from the band edge to ~ $8k_BT$ (~0.21 eV at 300 K) below for the $x = 0.12$ sample, which is exactly at the top of the first DOS peak. For $\mu_H$, the pristine sample shows a value of ~10 cm$^2$ V$^{-1}$ s$^{-1}$ due to its low $p$. The $\mu_H$ values for Pb-doped samples do not vary significantly, and are mainly in the range of 3–5 cm$^2$ V$^{-1}$ s$^{-1}$, even with a large variation in $E_F$. The large decrease of $\mu_H$ with Pb doping is primarily due to the involvement of heavy band in the electrical conduction and partially to the increased ionized impurity scattering, as confirmed by the high-temperature Hall measurements (300–650 K, Fig. 2c, d). $\mu_H$ of the pristine sample approximately follows a $T^{-1.5}$ relation, characteristic of acoustic phonon scattering (Fig. 2d). The temperature exponent of $\mu_H$ for the Pb-doped samples, however, evolves gradually from −1.5 to −0.5 with the increasing Pb content, primarily due to the increased ionized impurity scattering. The inter-valley scattering or increased carrier degeneracy with down-shifting the $E_F$ and increasing the conduction valleys may also partially contribute to this[29]. Moreover, as shown in Fig. 2c, except for the pristine sample displaying intrinsic conduction, the doped samples show nearly temperature-independent $p$, typical for

heavily doped semiconductors. Due to the rapid rise of DOS and $N_v$, we expect an unusual variation of electrical transport properties with down-shifting the $E_F$, which will be discussed in more detail below.

The PFs and ZTs are plotted as functions of Pb content at different temperatures (300, 473, 673, and 873 K), shown in Fig. 2e, f. The temperature dependence of electrical properties is shown in Supplementary Fig. 4. The maximum PF reaches ~11 µW cm$^{-1}$ K$^{-2}$, comparable to the best values in this material[10]. In particular, three PF maxima can be clearly observed with the increasing Pb content, corresponding to those of 4, 10, and 14 at%. For typical TE semiconductors with single parabolic band (SPB), there is usually one maximum at the optimal doping content and thus $E_F$, due to the interplay between $\sigma$ and $S$[30,31]. The anomalous three PF maxima should be related to the complex band structure or other extrinsic factors. Consequently, three ZT maxima can be observed for all temperatures, and at 873 K these values are ~0.9 ($x = 0.04$), ~1.1 ($x = 0.10$), and ~1.3 ($x = 0.14$). Clearly, the first two peaks at $x = 0.04$ and 0.10, within the Pb-doping limit, should be originated from the optimization of $p$ and/or the complex electronic band structure, such as the DOS peak at ~−0.2 eV and high $N_v$. The third peak, as will be demonstrated below, is ascribed to the compositing effect of PbSe in $Bi_{0.88}Pb_{0.12}CuSeO$.

To clarify the origin of the multiple PF peaks, the electrical transport properties of $Bi_{1-x}Pb_xCuSeO$ were modeled based on the Boltzmann transport theory under the relaxation-time approximation. Deformation potential coefficient, which represents the change in energy of the electronic band with elastic deformation and thus the coupling between electrons and phonons[32], is an intrinsic material parameter and was set as ~25 eV (normally ranges from 5 to 35 eV for semiconductors[33]). The modeling details can be found elsewhere[34], and the related parameters are listed in Supplementary Table 1. The effective masses ($m_d^*$) used for modeling were derived from the DFT calculated band structure and DOS, using the equation

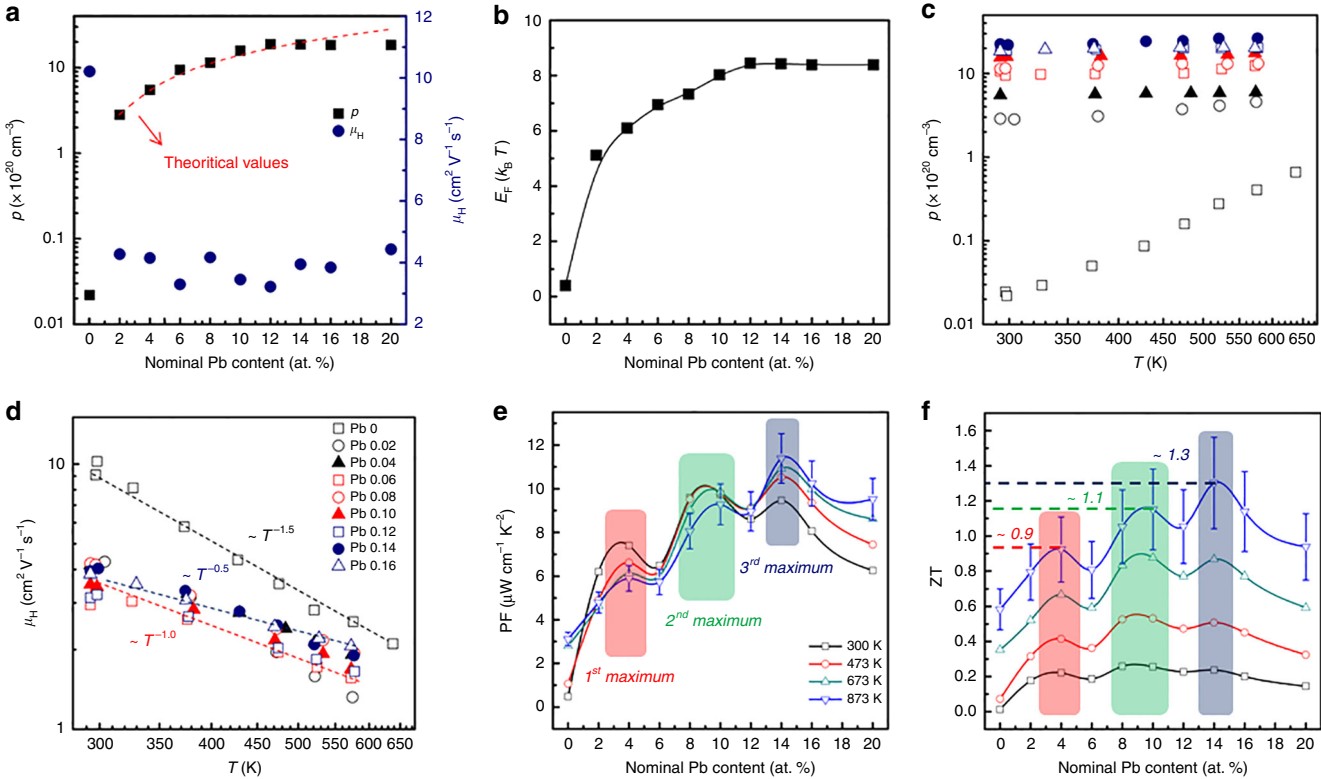

**Fig. 2** Electrical transport properties of $Bi_{1-x}Pb_xCuSeO$ ($x = 0$-$0.2$). **a** Carrier concentration $p$ and Hall mobility $\mu_H$ at room temperature. **b** Calculated Fermi energy ($E_F$) based on experimental $p$ and the calculated density of states (DOS). Temperature dependences of **c** $p$ and **d** $\mu_H$ (300–650 K). **e** Power factor (PF) and **f** dimensionless figure of merit (ZT) as a function of the nominal Pb content. Three maxima are marked as shaded areas for PF and ZT. The uncertainties of the Seebeck coefficient and the electrical conductivity measurements are ~3% and ~5%, respectively, giving rise to a PF error of ~10%. Combined with ~10% uncertainty in the thermal conductivity measurement, the measurement error of ZT is estimated to be ~20%

**Table 1 Related physical properties of $Bi_{1-x}Pb_xCuSeO$ ($x = 0$-$0.2$)**

| Samples | $\rho$ (g cm$^{-3}$) | $p$ ($10^{20}$ cm$^{-3}$) | $\mu_H$ (cm$^2$ V$^{-1}$ s$^{-1}$) | $S$ (μV K$^{-1}$) | $m^\star$ ($m_0$) | $L$ ($10^{-8}$ V$^2$ K$^{-2}$) | $N_v$ | PF (μW cm$^{-1}$ K$^{-2}$) | ZT |
|---|---|---|---|---|---|---|---|---|---|
| $x = 0$ | 8.74 | 0.02 | 10.2 | 367 | 0.8 | 1.50 | 1 | 3.1 | 0.6 |
| $x = 0.02$ | 8.72 | 2.8 | 4.3 | 179 | 3.3 | 1.66 | 8 | 4.8 | 0.8 |
| $x = 0.04$ | 8.68 | 5.5 | 4.2 | 142 | 5.4 | 1.75 | 11 | 5.9 | 0.9 |
| $x = 0.06$ | 8.49 | 9.5 | 3.3 | 114 | 5.6 | 1.86 | 13 | 5.7 | 0.8 |
| $x = 0.08$ | 8.64 | 11.5 | 4.2 | 117 | 6.5 | 1.84 | 14 | 8.1 | 1.0 |
| $x = 0.10$ | 8.38 | 15.9 | 3.5 | 110 | 6.7 | 1.87 | 14 | 9.3 | 1.1 |
| $x = 0.12$ | 8.37 | 18.9 | 3.2 | 98 | 6.8 | 1.92 | 14 | 9.0 | 1.0 |
| $x = 0.14$ | 8.23 | 18.7 | 3.9 | 93 | 7.0 | 1.96 | – | 11.4 | 1.3 |
| $x = 0.16$ | 8.21 | 18.5 | 3.8 | 88 | 6.6 | 1.98 | – | 10.2 | 1.1 |
| $x = 0.20$ | 8.16 | 18.5 | 4.4 | 72 | 5.3 | 2.06 | – | 9.5 | 0.9 |

The mass density ($\rho$), hole concentration ($p$), Hall mobility ($\mu_H$), Seebeck coefficient ($S$), density of states (DOS) effective mass ($m^\star$), Lorenz constant ($L$), band degeneracy ($N_v$) near the Fermi energy ($E_F$) at 300 K, and the power factor (PF) and dimensionless figure of merit (ZT) values at 873 K

$\phi(\varepsilon) = \frac{4\pi(2m_d^*)^{3/2}}{h^3}\varepsilon^{1/2}$, where $\varepsilon$ is the carrier energy, $\phi(\varepsilon)$ the DOS per unit volume, and $h$ the Planck constant. Rigid band model was assumed while deriving the $E_F$-dependent $m_d^\star$. The calculated results are plotted as blue and red lines in Fig. 3c for the pristine and 12.5 at% Pb-doped samples, respectively. The band structure and DOS of $Bi_{0.875}Pb_{0.125}CuSeO$, shown in Fig. 3a, were calculated for deriving the red line in Fig. 3c. We observe much higher $m^\star \propto p$ data, and larger DOS near the VBM (Fig. 3b) for Pb-doped samples as compared with those of the pristine BiCuSeO, where $m^\star$ is the DOS effective mass. As a result, the calculated data (red line in Fig. 3c) for $Bi_{0.875}Pb_{0.125}CuSeO$ are much higher than those of the pristine BiCuSeO (blue line),

well consistent with the experimental data of Pb-doped samples[23,26–28]. Here the experimental $m^\star$ values are calculated based on the experimental $S$ and $p$, with the assumption of acoustic phonon scattering and SPB model. The details can be found in Supplementary Note 2 or elsewhere[23]. The estimated experimental $m^\star$, shown in Table 1, increases rapidly from ~$0.8m_0$ ($m_0$ is the free electron mass) for the $x = 0$ sample to ~$7.0m_0$ for $x = 0.14$, indicating a significant band nonparabolicity. The band nonparabolicity of BiCuSeO has been widely reported, which is typical for narrow gap semiconductors[35]. Considering the minor contribution from Pb to the DOS of $Bi_{0.875}Pb_{0.125}CuSeO$ at the top of valence band, shown in Supplementary Fig. 5, the origin of large $m^\star$ for the Pb-doped

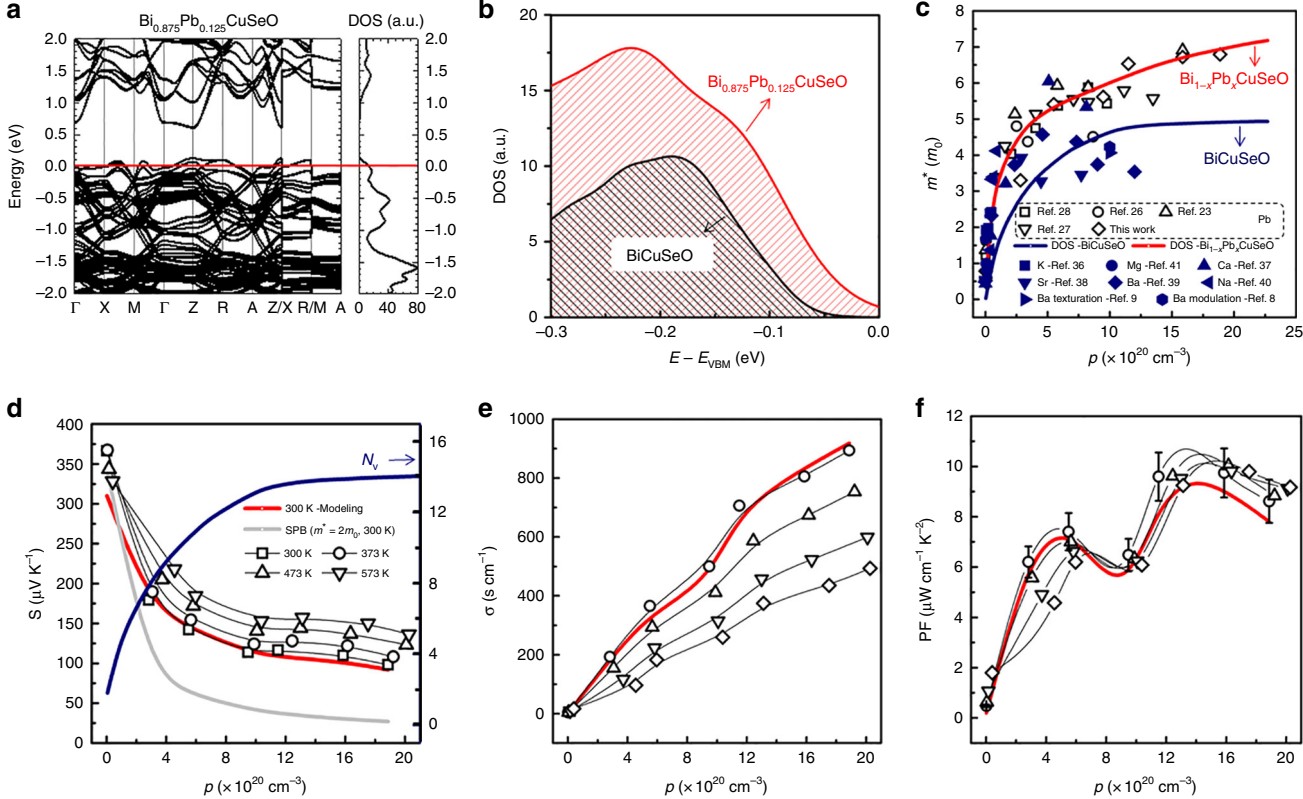

**Fig. 3** Electronic structure and calculated electrical transport properties of $Bi_{1-x}Pb_xCuSeO$. **a** Electronic band structure and total density of states (DOS) of $Bi_{0.875}Pb_{0.125}CuSeO$. **b** Comparison of DOS at valence band top (−0.3 to 0 eV) between BiCuSeO and $Bi_{0.875}Pb_{0.125}CuSeO$. Carrier concentration dependences of **c** the effective mass, **d** Seebeck coefficient, **e** electrical conductivity, and **f** power factor for $Bi_{1-x}Pb_xCuSeO$. The blue and red lines in **c** are estimated from the DOS of the pristine and Pb-doped BiCuSeO, respectively. The red lines in **d**–**f** are calculated based on the Boltzmann transport theory and calculated effective masses $m_d^*$ (the red line in **c**) of $Bi_{0.875}Pb_{0.125}CuSeO$ by assuming acoustic phonon scattering (300 K). The blue line in **d** is the estimated band degeneracy, and the gray line is estimated by assuming the SPB model and a constant effective mass of $2m_0$. The error bar of PF in **f** is 10%

samples is unclear and possibly related to the subtle change of chemical environments in the insulating $(Bi_2O_2)^{2+}$ layer and subsequently the conductive $(Cu_2Se_2)^{2-}$ layer. In addition, the calculated $m^*$ values of various alkaline or alkaline earth-doped samples (on the Bi site)[8,9,23,26–28,36–41], shown as the solid symbols in Fig. 3c, are apparently lower than those of Pb-doped samples, especially at high doping levels. $m^*$ values of non-Pb-doped samples are approximately consistent with the calculated line of the pristine BiCuSeO. In light of the unique role of Pb doping, the $m_d^*$ values of the Pb-doped samples were used for the following modeling.

As shown in Fig. 3d, e, the modeled $S$ and $\sigma$ values at 300 K agree well with the experimental data, indicating the validity of our model. Seebeck coefficient estimated from the SPB model ($m^* = 2m_0$), shown as the gray line in Fig. 3d, is obviously lower than the experimental data. More importantly, our modeling well reproduces the two peaks in PF, and clarifies that the two PF maxima can be ascribed to the rapid rise in DOS due to the convergence of multiple bands and the conventional optimization of $E_F$. With down-shifting the $E_F$ into the valence band, the involvement of multiple heavy bands, that is, the increasing $N_v$ (Fig. 3d) in the electrical conduction largely increases DOS and thus $S$ without sacrificing $\mu_H$, shown in Figs 3b and 2a, resulting in the first PF maximum. Further shift of $E_F$ increases $p$ and thus $\sigma$, while maintaining a large $S$ due to the large band degeneracy, which gives rise to the second maximum, shown in Fig. 3f. According to our experimental data, band structure calculations, and numerical modeling, to fully utilize these beneficial band features in BiCuSeO, including

the mixing of light and heavy bands and high band degeneracy, one has to down-shift the $E_F$ and touch the first DOS peak ($p > 10^{21}$ cm$^{-3}$). Thereby, high PF values (~10 μW cm$^{-1}$ K$^{-2}$) can be achieved without using other approaches such as texturing, modulation doping, chemical bonding engineering, and so on, and Pb doping on the Bi site does play a unique role in adjusting the $E_F$ and fully utilizing the electronic band structure.

**Compositing effects of PbSe for the $x > 0.12$ samples.** The first two maxima in PF and ZT can be well understood as originating from the intrinsically complex band structure; however, the appearance of third peak at $x \sim 0.14$ is intriguing. The $x > 0.12$ samples can be viewed as composites with appreciable amounts of PbSe and trace $Cu_2Se_\delta$ embedded in the Pb-doped BiCuSeO matrix with unchanged $p$ and $E_F$ values, and generally the compositing could trigger unexpected effects on electron and phonon transport, such as carrier-filtering and phonon-blocking[42]. By scrutinizing the transport data, we can find that further increase in PF for the $x > 0.12$ samples predominantly results from increases in $\mu_H$ and thus $\sigma$. This $\mu_H$ increase is primarily attributed to the appearance of PbSe precipitates with high hole mobility (~1000 cm$^2$ V$^{-1}$ s$^{-1}$ for lightly doped p-type PbSe at 300 K)[43,44], rather than $Cu_2Se_\delta$ nanodots with small volume fraction and low mobility (~11.1 cm$^2$ V$^{-1}$ s$^{-1}$ for the SHS-SPSed $Cu_2Se$ at 300 K)[22].

To confirm our speculation, the distributions of PbSe and $Cu_2Se_\delta$ were investigated by a scanning electron microscope

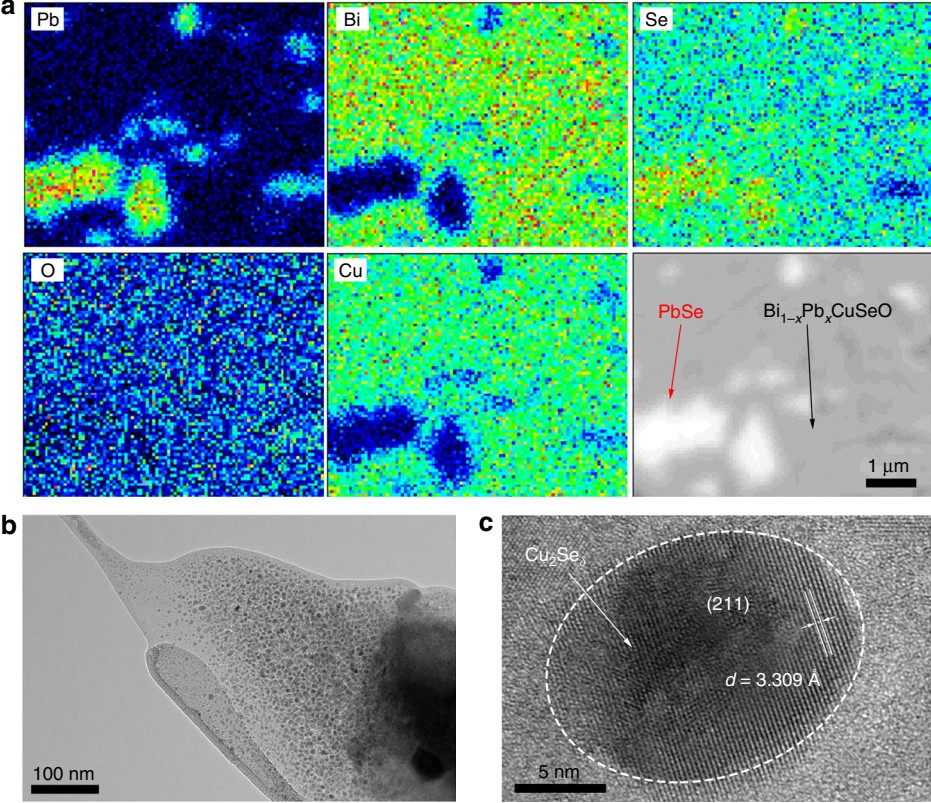

**Fig. 4** Microstructural and compositional characterizations of $Bi_{0.86}Pb_{0.14}CuSeO$. **a** Elemental mapping of Pb, Bi, Se, O, and Cu. The results clearly show the PbSe precipitates (white) embedded in $Bi_{1-x}Pb_xCuSeO$ matrix (gray). **b** Transmission electron microscope (TEM) image showing the homogeneously distributed nanodots. **c** High-resolution TEM (HRTEM) image confirming that the nanodots are $Cu_2Se_\delta$

(SEM), an electron probe micro-analyzer (EPMA), and a transmission electron microscope (TEM), as shown in Fig. 4 and Supplementary Fig. 6. There is no detectable impurity phase in SEM for the $x \le 0.12$ samples (Supplementary Fig. 7a), in agreement with the XRD and Hall data. Besides, the average grain size is largely decreased for the $x > 0.12$ samples (Supplementary Fig. 7b), probably due to the "pinning" effect introduced by the nano-precipitates and other secondary phases[45,46]. Uniformly distributed $Cu_2Se_\delta$ nanodots with sizes of 5–10 nm (confirmed by the inter-planar spacing of 3.309 Å, corresponding to the (211) plane, PDF #47-1448)[47] can be easily observed in the SHS-SPSed samples, shown in Fig. 4b, c, which is well consistent with our previous work[11]. The $Cu_2Se_\delta$ nanodots are primarily generated during the rapid and non-equilibrium SHS processes, and their amount and size both increase significantly for the $x > 0.12$ samples due to the non-stoichiometry, making them detectable in XRD (Supplementary Fig. 3a). In addition, PbSe precipitates with micrometer size can be readily observed in the elemental mapping (EPMA for well-polished samples), shown in Fig. 4a. Our quantitative EPMA analysis (averaged among 10 arbitrarily selected points) confirms that in $Bi_{0.86}Pb_{0.14}CuSeO$ there exist a large amount of p-type PbSe precipitates embedded in the $Bi_{1-x}Pb_xCuSeO$ ($x \sim 0.12$) matrix, consistent with the XRD and SEM/EDS results (Supplementary Fig. 6). In the $Bi_{1-x}Pb_xCuSeO$ ($x > 0.12$) samples, in situ formed PbSe could noticeably alter the transport properties through the compositing effect. Without detailed transport properties of the PbSe precipitates, it is hard to use the Bergman composite theory to quantitatively calculate the transport properties of the composites. Additional experimental efforts need to be carried out to fully corroborate the compositing effect, which are beyond the scope of the present study.

**Thermal transport properties**. In addition to the complex band structure, BiCuSeO is also well known for its glass-like thermal conduction. Figure 5a shows the temperature-dependent lattice thermal conductivity. $\kappa_L$ values for all samples decrease with the increasing temperature, approximately following a $T^{-1}$ relation and thus typical of the Umklapp scattering of phonons. Moreover, Pb doping does significantly reduce $\kappa_L$ in the whole temperature range, shown in Fig. 5b. Using the Debye–Callaway model[48,49], we estimated the effect of Pb doping on the $\kappa_L$ at 300 and 873 K by assuming the successful replacement of Pb on the Bi site, shown as the solid lines in Fig. 5b. The calculation details can be found in Supplementary Note 3 or elsewhere[23,50]. Here we did not consider the effect of oxygen vacancies that presumably exist in all samples with a negligible difference[51], nor the influence of $Cu_2Se_\delta$ nanodots. This assumption is valid for the $x \le 0.12$ samples, validated by the good agreement between the model and experimental data. The model overestimates $\kappa_L$ for $x > 0.12$, primarily due to the increased amounts of secondary phases, for example, $Cu_2Se_\delta$ nanodots, as compared with the pristine sample. Considering the small mass difference between Bi and Pb, the strong point defect scattering for phonons is mainly originated from the strain field fluctuation ($\Gamma_{SF}$, considering the ionic radius for $Bi-r_{Bi}^{3+} = 1.17$ Å and $Pb-r_{Pb}^{2+} = 1.29$ Å), as indicated by the dominated role of $\Gamma_{SF}$ in disorder scattering parameter $\Gamma$ (Supplementary Fig. 8). In addition to the strong point defect scattering for short-wavelength phonons, mesoscale grain boundaries, and nanostructures further strongly scatter the long- and middle-wavelength phonons, creating a hierarchically structural feature for scattering a wide spectrum of lattice phonons. Thereby, as shown in Fig. 5a, $\kappa_L$ is well below the estimated minimum value ($\kappa_{min} \sim 0.59$ W $m^{-1}$ $K^{-1}$, Cahill's glassy limit)[23], especially at high temperatures, indicating a substantial phonon scattering in the

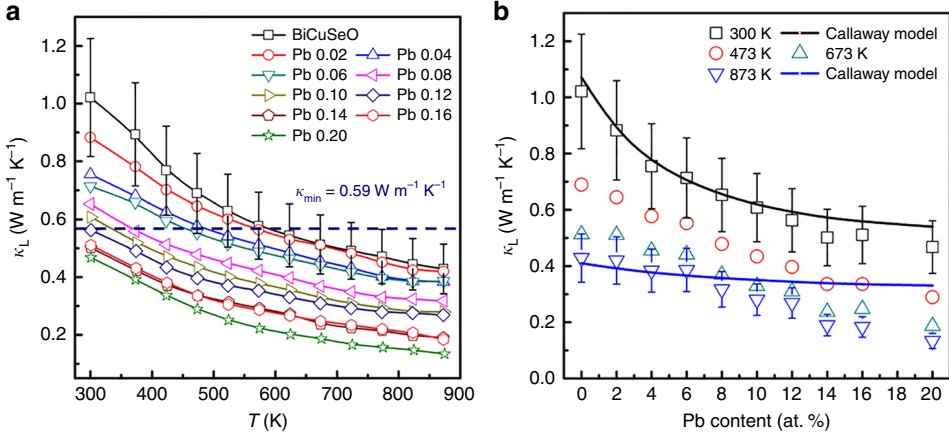

**Fig. 5** Thermal transport properties of $Bi_{1-x}Pb_xCuSeO$. **a** Temperature-dependent lattice thermal conductivity $\kappa_L$. **b** Pb-content-dependent $\kappa_L$ at 300, 473, 673, and 873 K. The solid lines in **b** were estimated by the Debye–Callaway model for 300 and 873 K. The error bars in **a**, **b** are both 20%, as the uncertainties of the total and electronic thermal conductivity are both estimated to be ~10%

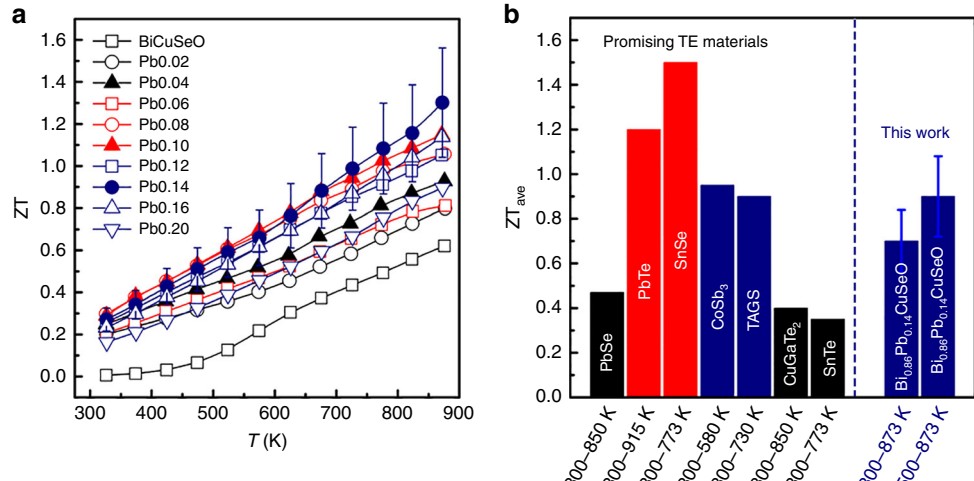

**Fig. 6** The dimensionless figure of merit ZT of $Bi_{1-x}Pb_xCuSeO$ (x = 0-0.2). **a** Temperature dependence of ZT. **b** Comparison of the average ZTs between $Bi_{0.86}Pb_{0.14}CuSeO$ and other materials reported in the literature[13,14,42,53–56]. The error bars in **a**, **b** are both 20%

SHS-SPSed $Bi_{1-x}Pb_xCuSeO$. The lowest $\kappa_L$ of ~0.13 W m$^{-1}$ K$^{-1}$ at 873 K can be achieved for $Bi_{0.8}Pb_{0.2}CuSeO$, which, to our best knowledge, is the lowest reported value in BiCuSeO. The ultralow $\kappa_L$ of $Bi_{1-x}Pb_xCuSeO$, much lower than the estimated $\kappa_{min}$, is presumably due to the decreased sound velocity upon doping, especially at elevated temperatures.

**Dimensionless figure of merit**. Due to the complex band structure and high band convergence of BiCuSeO, full exploration of the favorable band features leads to high TE figure of merits, in combination with the ultralow $\kappa_L$ originated from the hierarchical structural features. Figure 6a shows the temperature dependences of ZT for $Bi_{1-x}Pb_xCuSeO$, and three ZT maxima of 0.9, 1.1, and 1.3 at 873 K can be found in the x = 0.04, 0.10, and 0.14 samples, respectively. Based on first-principles calculations, numerical modeling, and experimental data, these maxima can be attributed to the convergence of multiple valence bands, the conventional optimization of the $E_F$, as well as the effects of compositing with p-type PbSe. The anomalous multiple maxima in TE performance with shifting $E_F$, in sharp contrast to most TE semiconductors with a single maximum, may provide a new perspective to explore the doping effects for other TE materials with complex band

structures, such as chalcogenides, skutterudites, and so on, in addition to the simple doping efficiency difference based on atom size, mass, and valence shell[52]. Furthermore, excessive doping and the introduction of in situ formed precipitates could be a viable strategy to produce nanocomposites with compositing effects or intensified phonon scattering.

In addition to the high peak ZTs, the average ZT values have also been calculated as 0.7 for 300–873 K and 0.9 for 500–873 K in $Bi_{0.86}Pb_{0.14}CuSeO$. As shown in Fig. 6b, these values are comparable to those of the best mid-temperature p-type TE materials[13,14,43,53–56], such as $CoSb_3$ and PbTe. These high ZTs, combined with the good thermal and chemical stabilities (shown in Supplementary Fig. 9), as well as the time- and cost-efficient SHS-SPS technique, make Pb-doped BiCuSeO a promising material for intermediate temperature power generation applications.

## Discussion

In this study, the electronic band structure and TE transport properties of BiCuSeO were systematically investigated to fully utilize the favorable band features. Pb-doped BiCuSeO samples (0–20 at%) were synthesized by an SHS-SPS technique. When

shifting the Fermi energy into the valence band, convergence of multiple valence bands largely increases the density of states effective mass without scarifying the carrier mobility, and thus results in the first PF maximum at a low hole concentration of $\sim 5.5 \times 10^{20}$ cm$^{-3}$. Further moving the Fermi energy deep into the valence band gives rise to the second peak in PF through balancing the electrical conductivity and Seebeck coefficient. In addition, in situ formed PbSe precipitates when the Pb content exceeds the solubility limit ($x > 0.12$) lead to the third PF peak due to the compositing effect. Consequently, three ZT peaks of 0.9, 1.1, and 1.3 at 873 K are observed for 4, 10, and 14 at% Pb-doped BiCuSeO samples, respectively, indicating the complex electronic structure and multiple roles of Pb in BiCuSeO. Meanwhile, the high average ZTs of 0.7 for 300–873 K and 0.9 for 500–873 K, making these materials promising for large-scale mid-temperature power generation applications. Specifically, our study demonstrates that, for material systems with complex band structures (nonparabolicity, multiple bands with varying band mass, etc.), rational tuning of the Fermi energy and dopant content within or exceeding the solubility limit may provide new opportunities to optimize the TE performance.

## Methods

**Material synthesis.** Polycrystalline Bi$_{1-x}$Pb$_x$CuSeO ($x = 0, 0.02, …, 0.2$) samples were prepared by a self-propagating high-temperature synthesis and SHS-SPS technique. Stoichiometric amounts of Bi (99.99%, Aladdin), Bi$_2$O$_3$ (99.99%, Aladdin), PbO (99.9%, Aladdin), Cu (99.99%, Aladdin), and Se (99.99%, Aladdin) were mixed by hand grinding. The mixed powders were cold pressed into pellets and underwent the SHS processes. The details of the SHS method can be found elsewhere[23]. Given that a significant volatilization of Se may exist during the SHS processes, additional 5 at% Se was added. The obtained SHSed powders were then compacted into dense pellets with a diameter of ~12.5 mm and a height of ~10 mm by SPS at 923 K under a uniaxial pressure of 50 MPa for 5 min.

**Structural characterization.** Phase purity and crystal structure were investigated by powder XRD (Bruker D8 Advance, Germany). Chemical uniformities and compositions of the samples were characterized by an EPMA (JXA-8230, JEOL, Japan). The chemical compositions were averaged from 10 arbitrarily selected points. The microstructures of the samples with different compositions were examined by a field-emission SEM (MERLIN Compact FE-SEM, Carl Zeiss, Germany) and a high-resolution transmission electron microscope (JEOL2010, Japan).

**Transport property measurement.** The sintered pellets were cut into $10 \times 10 \times 1.5$ mm$^3$ thin square sheets and $3 \times 3 \times 12$ mm$^3$ bar-shaped specimens, and all transport properties were measured perpendicular to the SPS pressure direction. The rectangular bars were used for the simultaneous measurements of $S$ and $\sigma$ from room temperature to 873 K using a ZEM-3 (ULVAC, Japan). Hall coefficient measurements (300–650 K) were performed on a homemade system equipped with a 2 T electromagnet, and the magnetic field is determined by a flux meter. A four-probe configuration and pressure contact were used for the measurements. The carrier concentration ($p$) and Hall mobility ($\mu_H$) were estimated from the measured Hall coefficient ($R_H$) and electrical conductivity by the relations $p = r/e|R_H|$ (assuming the Hall factor $r = 1.0$) and $\mu_H = \sigma|R_H|$. Thermal diffusivity ($D$) was measured by a laser flash method (LFA-457, Netzsch, Germany) under a continuous Ar flow, shown in Supplementary Fig. 1a. The total thermal conductivity ($\kappa$) was calculated by using $\kappa = DC_P\rho$, where $\rho$ is the mass density measured by the Archimedes method (Table 1) and $C_p$ specific heat calculated by the Debye model[57–59] (similar to the Dulong–Petit law at high temperatures, Supplementary Fig. 1b and Note 1). The uncertainties of the Seebeck coefficient and the electrical conductivity measurements are estimated to be ~3% and ~5%, respectively. The uncertainty in the thermal conductivity measurement is estimated within 10%. As a result, the uncertainty of the figure of merit ZT is accumulated to be ~20%.

**Computational technique.** The electronic structures were calculated using the projector augmented wave method, as implemented in the Vienna ab initio Simulation Package[60,61]. The mBJ exchange potential[62] and the generalized gradient approximation[63] were used in the calculations. To mimic the Pb doping, we constructed a $2 \times 2 \times 1$ supercell of the primitive cell. Both supercells with or without the Pb dopants were calculated. The lattice parameters and ionic positions were fully relaxed. SOC was considered in the band structure calculations.

## Data availability
The authors declare that all data supporting the findings of this study are available within the paper and its supplementary information, or from the authors upon reasonable request.

## Code availability
The code that support the findings of this study are available from the corresponding author upon reasonable request.

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

## Acknowledgements

This work was financially supported by the National Key Research Program of China, under grant no. 2016YFA0201003, Sichuan Science and Technology Program under grant no. 2019JDJQ0055, the Ministry of Science and Technology of China through a 973-Project, under grant no. 2013CB632506, NSF of China under grant nos. 51672155 and 51202232, and by the National Science Foundation of U.S. under award no. 1235535.

## Author contributions

G.-K.R., Y.H.L., and J.Y. proposed the research. G.-K.R., S.W., and J.Y. designed the experiments. G.-K.R. and Z.Z. performed the material synthesis, characterization, measurements of electrical and thermal properties, and analyzed the data. X.L., J.Y., and W.Z. performed the first-principle calculations. G.-K.R., S.W., Y.L., J.Y., and C.-W.N. co-wrote the paper. All authors discussed the results and commented on the manuscript.

## Additional information

**Competing interests:** The authors declare no competing interests.

