## [Peer Review File · Nature Communications]

Reviewers' comments:

Reviewer #1 (Remarks to the Author):

The authors investigated the thermoelectric properties of BiCuSeO, and showed that Pb-doping would introduce three power factor maxima rather than typical one upon carrier concentration optimization for most TE semiconductors, which can be further attributed the Fermi energy optimization, band convergence, and compositing effects due to in-situ formed PbSe precipitates based on sufficient analysis. This work established an accurate band structure characterization for BiCuSeO and identify the role of band convergence and nanoprecipitation as the driving mechanism for achieving high ZT, further three peaks of 0.9, 1.1, and 1.3 at 873 K have been achieved for 4 at.%, 10 at.%, and 14 at.% Pb-doped samples, respectively. It is an innovative work and the paper is well written, this paper is publishable in Nature Communications after addressing follow comments:

1) The results indicated that Pb-doping could enhance the performance of BiCuSeO with these new mechanism, how about other elemental doping (Ba, Sr, Ca, Mg)? The author should explain more about this since the title is "High Thermoelectric Performance in BiCuSeO: Complex Band Structure and Beyond".

2) For the "Introduction" part and following sections, the authors have demonstrated the band structure of BiCuSeO is nonparabolic, and many conduction valleys have given rise to the electrical transport. Nevertheless, a single parabolic band model has been used to estimate the dependence of the effective mass and Lorenz number on the concentration of carriers, which is confusing.

3) In Fig. S3a and Fig. 4, the evidence of XRD and TEM both indicated that other than PbSe precipitates, Cu₂Seδ nanodots are also formed. For the third maximum of PF, the authors have attributed it to the appearance of PbSe, how about the increasing amount of Cu₂Seδ? Will it play more important role than that of PbSe for this compositing system? A little more analysis of the comparison between two impurities that cause the third PF/ZT maximum may be helpful.

4) As shown in Fig.2, the "Pb content" is actual or nominal? Please address and mark it on the X axis.

5) Fig 1 shows the electronic band and the Fermi surface, while the corresponding energy level of each Fermi surface was not present. Only three carrier concentrations and corresponding Fermi surfaces selected, which cannot fully cover the values listed in Table.1 with the respect of N_v. Authors should present the Fermi surface with typical N_v and corresponding energy level in band structure.

6) BiCuSeO has strong anisotropy, especially in polycrystalline. In calculation, I am wondering whether the anisotropy was considered in electrical conductivity (Fig. 3e) and lattice thermal conduction (Fig. 5b) estimation?

7) When authors mentioned BiCuSeO, the first report should be cited: Zhao et al. Applied Physics Letters, 97 (2010) 092118.

Reviewer #2 (Remarks to the Author):

BiCuSeO has received more and more attentions as a high performance thermoelectric materials. As usual, the thermoelectric materials only have one power factor maximum upon carrier concentration optimization. Interestingly, the authors reported a novel phenomenon in BiCuSeO system that the three power factor maxima were observed when doping Bi with Pb. Based on DFT calculations, numerical modeling, and experimental investigation, the three maxima were attributed to the Fermi energy optimization, band convergence, and compositing effect due to in-situ formed PbSe precipitates. The experiments were well designed and elaborately conducted, and the results are interesting and reasonably discussed. This work is important for the thermoelectric community and of high interest to the materials scientists. In general, I feel that this manuscript should be accepted for publication in Nat. Comm. with a minor revision. Please response to the following questions during revision:

1. Why the calculated C_p was used for calculating thermal conductivity?
2. In supporting information, XRD patterns, there is an impurity peak around 40° for the pristine sample and $Pb_{0.08}$ sample, but without any explanation and indexation.
3. Fig. 4 a, the elemental mapping, the makers of Bi, Se, O, Cu are hard to be seen.
4. The trace of Cu_2Se was observed. Is it caused by Pb doping or intrinsic in BCSO system?
5. Is the sample thermally stable? The repeat of thermoelectric properties measurement is required.

Responses to the referees' comments

Referee #1:

Comments to the Author

The authors investigated the thermoelectric properties of BiCuSeO, and showed that Pb-doping would introduce three power factor maxima rather than typical one upon carrier concentration optimization for most TE semiconductors, which can be further attributed the Fermi energy optimization, band convergence, and compositing effects due to *in-situ* formed PbSe precipitates based on sufficient analysis. This work established an accurate band structure characterization for BiCuSeO and identify the role of band convergence and nanoprecipitation as the driving mechanism for achieving high ZT , further three peaks of 0.9, 1.1, and 1.3 at 873 K have been achieved for 4 at.%, 10 at.%, and 14 at.%

Pb-doped samples, respectively. It is an innovative work and the paper is well written, this paper is publishable in *Nature Communications* after addressing follow comments.

Response: We appreciate referee's positive assessment of our manuscript as being an "innovative work and the paper is well written".

- 1) The results indicated that Pb-doping could enhance the performance of BiCuSeO with these new mechanism, how about other elemental doping (Ba, Sr, Ca, Mg)? The author should explain more about this since the title is "High Thermoelectric Performance in BiCuSeO: Complex Band Structure and Beyond".

Response: Like other TE materials in which the single parabolic band (SPB) model applies, it is noted that elemental doping, such as using Pb, Ba, Sr, Ca and Mg, mainly optimizes the carrier concentration of BiCuSeO, hence typically resulting in one single PF maximum. This has been widely reported in the literature, even when a strong band nonparabolicity is observed. These results, nevertheless, may be primarily attributed to their relatively small carrier concentrations due to low solubility limits (e.g., Sr and Mg) [*Appl. Phys. Lett.*, 2010, **97**, 092118; *J. Alloy Compd.*, 2013, **551**, 649-653], or to the perception that only one PF (*ZT*) maximum exists in TE materials.

In fact, as we have demonstrated in this work, the first two PF and *ZT* maxima are intrinsic to the BiCuSeO band structure, and other elemental doping (such as Ba, Sr, Ca, Mg) could also introduce more than one PF and/or *ZT* maxima at sufficiently high carrier concentrations. As shown in Fig. 3f, our computational results based on a rigid band model have clearly shown two PF maxima in the entire carrier concentration range, consolidating the complex band structure of BiCuSeO. Two PF maxima in Ba-doped BiCuSeO with varying the carrier concentration and temperature [*Energy Environ. Sci.*, 2012, **5**, 8543-8547] has been reported previously. In this work, three PF maxima rather than one can be achieved for the Pb-doped BiCuSeO, which can be further attributed to

the Fermi energy optimization, band convergence, and compositing effects by *in-situ* forming PbSe precipitates.

- 2) For the “Introduction” part and following sections, the authors have demonstrated the band structure of BiCuSeO is nonparabolic, and many conduction valleys have given rise to the electrical transport. Nevertheless, a single parabolic band model has been used to estimate the dependence of the effective mass and Lorenz number on the concentration of carriers, which is confusing.

Response: Thanks for the comment. Indeed, for semiconductors with nonparabolic bands, using the single parabolic band (SPB) model could introduce some errors in calculating the band parameters, especially the Fermi energy. However, the calculations of density of states effective mass and Lorenz number, based on experimental Seebeck coefficient and carrier concentration, are sufficiently accurate. Thus, this model has been widely used for many narrow-gap semiconductors with a significant band nonparabolicity, *e.g.*, BiCuSeO, Bi₂Te₃, PbTe, CoSb₃, Mg₂Si, *etc.* The related references are *NPG Asia Mater.*, 2013, **5**, 47; *J. Mater. Chem.*, 2011, **21**, 12259-12266; *NPG Asia Mater.*, 2014, **6**, 88; *Energy Environ. Sci.*, 2013, **6**, 3346-3355; *J. Appl. Phys.*, 1996, **80**, 4442-4449; *etc.* Indeed, the increase of estimated effective mass with the increasing Fermi energy has been widely used to validate the band nonparabolicity. Therefore, in this study, we use the SPB to model the transport properties, such as the effective mass and the Lorenz number, *etc.*

We agree that considering the effects of nonparabolicity and multiband the obtained results are convoluted, however, the adjustable parameters related to nonparabolicity, band offset, *etc.* are typically difficult to be accurately determined. Especially when multiple bands are being considered, the calculations will be rather complex. This, we believe, is also beyond the scope of this paper, and will be studied in our future work. Thanks very much for your valuable comment!

3) In Fig. S3a and Fig. 4, the evidence of XRD and TEM both indicated that other than PbSe precipitates, $\text{Cu}_2\text{Se}_\delta$ nanodots are also formed. For the third maximum of PF, the authors have attributed it to the appearance of PbSe, how about the increasing amount of $\text{Cu}_2\text{Se}_\delta$? Will it play more important role than that of PbSe for this compositing system? A little more analysis of the comparison between two impurities that cause the third PF/*ZT* maximum may be helpful.

Response: This is a good point. $\text{Cu}_2\text{Se}_\delta$ nanodots, primarily generated during the rapid and non-equilibrium SHS processes and thus existing in all samples [*Energy Environ. Sci.*, 2017, **10**, 1590-1599; *RSC Adv.*, 2015, **5**, 69878-69885], are not responsible for the third PF or *ZT* peaks.

First of all, we didn't find any $\text{Cu}_2\text{Se}_\delta$ nanodots amount dependence on the Pb-content. Secondly, as shown in Fig. 2d, the hole mobility of the $x > 12$ at.% samples with PbSe precipitates, increases noticeably as compared with those of 10 at.% and 12 at.% Pb-doped BiCuSeO. This anomalous mobility increase is the primary origin of the third PF maximum, and can't be explained by the band structure and transport theory without considering the compositing effect of PbSe. The appearance of p-type PbSe with high carrier mobility ($\sim 1000 \text{ cm}^2 \text{ V}^{-1} \text{ s}^{-1}$ for lightly doped p-type PbSe at 300 K) [*Adv. Mater.*, 2011, **23**, 1366-1370] well accounts for the third PF maximum, while compositing with the $\text{Cu}_2\text{Se}_\delta$ nanodots with small hole mobility ($\sim 11.1 \text{ cm}^2 \text{ V}^{-1} \text{ s}^{-1}$ for SHS-SPSed Cu_2Se at 300K [*Nat. Commun.*, 2014, **5**, 4908]) can't give rise to increased hole mobility, as demonstrated in a recent work that no hole mobility increase was observed in BiCuSeO composited with a small amount of $\text{Cu}_2\text{Se}_\delta$ (< 15 wt.%) [*J. Alloy. Compd.*, 2016, **662**, 320-324]. Thirdly, $\text{Cu}_2\text{Se}_\delta$ nanodots with sizes of 5-10 nm should only have a negligible influence on hole transport, considering their small volume fraction. Though, the $\text{Cu}_2\text{Se}_\delta$ nanodots play an important role in phonon scattering and thus the lattice thermal conductivity reduction, which, combined with the third PF maximum resulted from PbSe precipitates, leads to the third *ZT* maximum. Therefore, we believe that $\text{Cu}_2\text{Se}_\delta$ nanodots

play a minor effect on the electrical transport and the third PF peak. To clarify these, we have modified the manuscript accordingly.

Page 10- “Compositing effects of PbSe for the $x > 0.12$ samples” section:

By scrutinizing the transport data, we find that further increase in PF for the $x > 0.12$ samples is predominantly resulted from the increases in μ_H and thus σ . This μ_H increase is primarily attributed to the appearance of PbSe precipitates with high hole mobility ($\sim 1000 \text{ cm}^2 \text{ V}^{-1} \text{ s}^{-1}$ for lightly doped p-type PbSe at 300 K^{43,44}), rather than the $\text{Cu}_2\text{Se}_\delta$ nanodots with small volume fraction and low mobility ($\sim 11.1 \text{ cm}^2 \text{ V}^{-1} \text{ s}^{-1}$ for the SHS-SPSed Cu_2Se at 300 K).²⁵

4) As shown in Fig.2, the “Pb content” is actual or nominal? Please address and mark it on the X axis.

Response: Thanks for the suggestion. In Fig. 2, all Pb contents shown are nominal, and thus we have changed all x-axis into “Nominal Pb content” in the revised manuscript per referee’s suggestion.

Page 7- “Exploring the Complex Band Structure” section:

Fig. 2 Electrical transport properties of $\text{Bi}_{1-x}\text{Pb}_x\text{CuSeO}$ ($x = 0-0.2$). **a** Carrier concentration p and Hall mobility μ_H at room temperature. **b** Calculated Fermi energy (E_F) based on experimental p and the calculated DOS. Temperature dependences of **c** p and **d** μ_H (300-650 K). **e** Power factor (PF) and **f** dimensionless figure of merit (ZT) as a function of the nominal Pb-content. Three maxima are marked as shaded areas for PF and ZT .

5) Fig 1 shows the electronic band and the Fermi surface, while the corresponding energy level of each Fermi surface was not present. Only three carrier concentrations and corresponding Fermi surfaces were selected, which cannot fully cover the values listed in Table.1 with the respect of N_v . Authors should present the Fermi surface with typical N_v and corresponding energy level in band structure.

Response: Thanks for the referee's suggestion. In this work, Fermi surfaces at various carrier concentrations shown in Fig. 1 are used to corroborate the band convergence due to multiple conduction valleys, and it is noted that even for the lightly-doped samples with E_F just below the VBM ($p = 3.0 \times 10^{20} \text{ cm}^{-3}$), the Fermi surface consists of multiple valleys rather than a single parabolic band, including dual-degenerated valleys at the Z

point, quadruplets along the Γ - M line, and eightfold-degenerated valleys along the Z - R line. Furthermore, down-shifting E_F simultaneously increases the contributions of these valleys and the carrier conduction. The trend doesn't change in the entire carrier concentration range studied and N_v maximum is 14. Therefore, three carrier concentrations with the corresponding Fermi surfaces should be sufficient to reveal this trend. Meanwhile, to compare band convergence among the BiCuSeO samples with various carrier concentrations, the N_v values listed in Table 1 are calculated using $m^* = N_v^{2/3} m_b^*$, rather than directly counting from the calculated Fermi surfaces, where the band effective mass m_b^* is estimated to be $0.8m_0$ for the undoped BiCuSeO ($N_v=1$).

- 6) BiCuSeO has strong anisotropy, especially in polycrystalline. In calculation, I am wondering whether the anisotropy was considered in electrical conductivity (Fig. 3e) and lattice thermal conduction (Fig. 5b) estimation?

Response: Thanks for the comment. As shown in the experimental section, “*The sintered pellets were cut into $10 \times 10 \times 1.5 \text{ mm}^3$ thin square sheets and $3 \times 3 \times 12 \text{ mm}^3$ bar-shaped specimens, and all transport properties were measured perpendicular to the SPS pressure direction*”. The electrical and thermal properties were measured along the in-plane direction to avoid the influence of microstructural anisotropy. Therefore, transport properties were measured in the same direction, and thus the obtained ZT s should be accurate without the influence of microstructural anisotropy.

- 7) When authors mentioned BiCuSeO, the first report should be cited: Zhao et al. Applied Physics Letters, 97 (2010) 092118.

Response: We thank the referee for the valuable suggestion and have cited the corresponding reference.

Referee #2:

Comments to the Author

BiCuSeO has received more and more attentions as a high-performance thermoelectric material. As usual, the thermoelectric materials only have one power factor maximum upon carrier concentration optimization. Interestingly, the authors reported a novel phenomenon in BiCuSeO system that the three power factor maxima were observed when doping Bi with Pb. Based on DFT calculations, numerical modeling, and experimental investigation, the three maxima were attributed to the Fermi energy optimization, band convergence, and compositing effect due to in-situ formed PbSe precipitates. The experiments were well designed and elaborately conducted, and the results are interesting and reasonably discussed. This work is important for the thermoelectric community and of high interest to the materials scientists. In general, I feel that this manuscript should be accepted for publication in Nat. Comm. with a minor revision.

Response: Thanks very much for the referee's comments on our work being "important for the thermoelectric community and of high interest to the materials scientists".

1. Why the calculated C_p was used for calculating thermal conductivity?

Response: This is a good question. To obtain the total thermal conductivity, the formula $\kappa = DC_p\rho$ is used, where thermal diffusion coefficient D was measured by a laser flash method (LFA-457, Netzsch, Germany) and the mass density ρ was measured by the Archimedes method. The heat capacity C_p is normally measured by DSC, however, the C_p of BiCuSeO measured by DSC shows a large variation with temperature or even negative temperature dependence, which would cause large errors in calculating thermal conductivity. This may be related to a slight sublimation of Se or minute endo- or exothermal reactions or processes upon heating-up.

In general, C_p (Fig. S1b) shows a weak temperature dependence and roughly equals to C_v (constant volume specific heat capacity) in solids at $T > \Theta_D$ (Debye temperature). Here using the Debye model (analogue to the Dulong–Petit law when $T > \Theta_D$) [*Phys. Rev. B* 2007, **75**, 125403; *Thermochim. Acta*, 1995, **269**, 109-116], we deduced C_p of BiCuSeO, which is close to previous reported C_p values of BiCuSeO ($\sim 0.268 \text{ J g}^{-1} \text{ K}^{-1}$ at 923K for pristine BiCuSeO [*NPG Asia Mater.*, 2013, **5**, 47]). This method for determining C_p values has also been widely used in thermoelectric community in many other TE materials, like the SnTe–AgSbTe₂ alloys, PbTe, PbSe, Mo₃Sb_{7-x}Te_x and Bi₂Te₃ etc. [*Adv. Energy Mater.*, 2012, **2**, 58-62; *Adv. Funct. Mater.*, 2011, **21**, 241-249; *Adv. Mater.*, 2011, **23**, 1366-1370; *J. Alloy Compd.*, 2007, **427**, 324-329; *Mater. Today Phys.*, 2017, **2**, 62-68; etc.]. This is primarily due to the difficulties in accurately measuring C_p using DSC.

2. In supporting information, XRD patterns, there is an impurity peak around 40° for the pristine sample and Pb0.08 sample, but without any explanation and indexation.

Response: Thanks for the question. The peak at 40.3 ° could be indexed to the (004) pattern of BiCuSeO according to PDF#45-0296, as shown in Fig. S3a, rather from an impurity phase. We have labeled all peaks in the XRD pattern for clarification.

3. Fig. 4 a, the elemental mapping, the makers of Bi, Se, O, Cu are hard to be seen.

Response: Thanks for the suggestion. We have optimized the figure in the revised manuscript, as shown below.

Page 11- “Compositing effects of PbSe for the $x > 0.12$ samples” section:

Fig. 4 Microstructure and composition characterizations of $\text{Bi}_{0.86}\text{Pb}_{0.14}\text{CuSeO}$. **a** Elemental mapping of Pb, Bi, Se, O, and Cu (EPMA). The results clearly show the PbSe precipitates (white) embedded in $\text{Bi}_{1-x}\text{Pb}_x\text{CuSeO}$ matrix (grey). **b** Transmission electron microscope (TEM) image showing the homogeneously distributed nanodots. **c** High resolution transmission electron microscope (HRTEM) image confirming that the nanodots are Cu_2Se_6 .

4. The trace of Cu_2Se was observed. Is it caused by Pb doping or intrinsic in BCSO system?

Response: This is a very good point. The Cu_2Se_6 nanodots are primarily generated during the rapid, non-equilibrium SHS processes, which have been systematically analyzed in our previous work [*Energy Environ. Sci.*, 2017, **10**, 1590-1599; *RSC Adv.*, 2015, **5**,

69878-69885]. Therefore, it is intrinsic only for BiCuSeO synthesized by the SHS-SPS method, and Pb-doping would not significantly affect the amount of $\text{Cu}_2\text{Se}_\delta$ nanodots. With the increasing Pb content over the solubility limit, the formation of PbSe precipitates will cause non-stoichiometry and thus indirectly increases the $\text{Cu}_2\text{Se}_\delta$ volume fraction, as shown in the XRD patterns in Fig. S3a.

5. Is the sample thermally stable? The repeat of thermoelectric properties measurement is required.

Response: Thank you for the good question. Thermal stability is certainly critical for practical applications. We have repeated the electrical property measurements for these samples, as shown in Fig. R1 and Fig. S9. It is obvious that our SHS-SPSed samples are thermally stable, at least in our repeatable measurements. However, more thorough and systematic studies, including long-time annealing in air or vacuum or heating-cooling cycling, are required to fully characterize the thermal stability of our SHS-SPSed samples. This is however beyond the scope of the present study and will be in our future work.

Fig. R1 Repeated measurements of the electrical properties (**a** electrical conductivity; **b** Seebeck coefficient; **c** power factor) for Bi_{1-x}Pb_xCuSeO samples ($x = 0-0.2$) at 300-875 K. The solid symbols are indexed to the initial results (same to those of Fig. S4), whereas the open symbols are the 2nd measurements.

Other revisions in the revised paper are shown as below:

- 1) In Page 1, we have added *Zhifang Zhou* from Tsinghua University as a co-author, who has provided experimental assistance, especially for the measurements of electrical properties in this work. Therefore, the “Author contributions” section has been modified accordingly.

Page 17- “Author contributions” section:

G.R., Y.L. and J.Y. proposed the research. G.R., S.W. and J.Y. designed the experiments. G.R. and Z.Z. performed the material synthesis, characterization, measurements of electrical and thermal properties, and analyzed the data. X.L., J.Y. and W.Z. performed the first-principle calculations. G.R., S.W., Y.L., J.Y. and C.N. co-wrote the paper. All authors discussed the results and commented on the manuscript.

- 2) In Page 16, we have added the funding supported by Sichuan Science and Technology Program, and modified the acknowledgements accordingly.

Page 16- “Acknowledgements” section:

This work was financially supported by the National Key Research Programme of China, under grant No. 2016YFA0201003, Sichuan Science and Technology Program under grant No. 2019JDJQ0055, the Ministry of Sci. & Tech. of China through a 973-Project, under grant No. 2013CB632506, NSF of China under grant No. 51672155 and 51202232, and by the National Science Foundation of U. S. under award No. 1235535.

- 3) In Page 16, the “Data availability” and “Code availability” sections have been added.

Page 16- “Data availability” section:

The authors declare that all data supporting the findings of this study are available within the paper and its supporting information, or from the authors upon reasonable request.

Page 16- “Code availability” section:

The code that support the findings of this study are available from the corresponding author upon reasonable request.

In summary, our study demonstrates that for material systems with complex band structures (nonparabolicity, multiple bands with varying band mass, *etc.*), rational tuning of the Fermi energy and dopant content within or exceeding the solubility limit may provide new opportunities to optimize their TE performance. The high average ZT s of 0.7 for 300-873 K and 0.9 for 500-873 K, together with the time- and cost-effective synthesis, makes SHS-SPSed Pb-doped BiCuSeO promising for large-scale mid-temperature power generation applications. Therefore, we believe our manuscript is valuable and will benefit the readership of *NC*, as the referees agreed. The manuscript has been revised together with the point-by-point responses. We look forward to publishing our manuscript in *NC* soon!

REVIEWERS' COMMENTS:

Reviewer #1 (Remarks to the Author):

authors addressed all my concerns, it can be published after addressing following point: "When authors mentioned BiCuSeO, the first report should be cited: Zhao et al. Applied Physics Letters, 97 (2010) 092118.", I think authors may misunderstood my comment in the first round, this paper firstly reported BCSO compound, therefore, this paper (ref. 7) should be cited earlier than references 5 and 6.

Reviewer #2 (Remarks to the Author):

The authors have responded all my comments and I am glad to agree to accept it.

May 13th, 2019

Editor
Nature Communications

Dear Editor,

Thank you for the acceptance of our manuscript titled “High Thermoelectric Performance in BiCuSeO: Complex Band Structure and Beyond” (Article ID: NCOMMS-18-26830C) submitted to *Nature Communications*. We appreciate the referees’ valuable and helpful comments. We have carefully considered their questions, provided point-to-point responses, and modified the manuscript accordingly. We believe that the revised manuscript has addressed all referees’ questions and hope that it could be published soon.

Responses to the referees’ comments

Referee #1:

Comments to the Author

Authors addressed all my concerns, it can be published after addressing following point: "When authors mentioned BiCuSeO, the first report should be cited: Zhao et al. Applied Physics Letters, 97 (2010) 092118.", I think authors may misunderstood my comment in the first round, this paper firstly reported BCSO compound, therefore, this paper (ref. 7) should be cited earlier than references 5 and 6.

Response: We thank the referee for the comment and have re-arranged the corresponding references in the revised version, as shown below.

Page 2- “Introduction” section:

With the virtues of ultralow lattice thermal conductivity ($\kappa_L \sim 0.4 \text{ W m}^{-1} \text{ K}^{-1}$ for the pristine sample at high temperatures), high Seebeck coefficient, and good thermochemical stabilities, BiCuSeO is one of the most promising candidates for mid-temperature TE power generation applications.⁵⁻⁷

Page 17- “References” section:

- 5 Zhao, L. D. *et al.* Bi_{1-x}Sr_xCuSeO oxyselenides as promising thermoelectric materials. *Appl. Phys. Lett.* **97**, 092118 (2010).

Referee #2:

Comments to the Author

The authors have responded all my comments and I am glad to agree to accept it.

Response: Thanks very much for the referee’s positive comment on our work.

Other revisions in the revised paper are shown as below:

1. On page 1, the sentence “The established literature on these materials shows only one power factor maximum upon carrier concentration optimization which is typical for most for the TE semiconductors” has been changed to “The established literature on these materials shows only one power factor maximum upon carrier concentration optimization, which is typical for most thermoelectric semiconductors”.
2. On page 3, the acronym “CBM” has been changed to “conduction band maximum (CBM)”.
3. On page 5, the sentence “The μ_H values for Pb-doped samples does not vary significantly, mainly in the range of $3\text{-}5 \text{ cm}^2 \text{ V}^{-1} \text{ s}^{-1}$ ” has been changed to “The μ_H

values for Pb-doped samples do not vary significantly, and are mainly in the range of $3\text{-}5\text{ cm}^2\text{ V}^{-1}\text{ s}^{-1}$.

4. On page 6, the sentence “High temperature PFs and ZT s are plotted as functions of Pb-content at different temperatures (300 K, 473 K, 673 K, and 873 K)” has been corrected to “The PFs and ZT s are plotted as functions of Pb-content at different temperatures (300 K, 473 K, 673 K, and 873 K)”.
5. Some words and/or typos have been corrected in the revised version, *e.g.*, on page 8, the “unfilled symbols” has been changed to “solid symbols”; on page 12, “low-wavelength phonons” has been changed to “short-wavelength phonons”.

The manuscript has been revised with the point-by-point responses. We look forward to publishing our manuscript in *NC* soon!